# Preparation of Ce–Mn Composite Oxides with Enhanced Catalytic Activity for Removal of Benzene through Oxalate Method

**DOI:** 10.3390/nano9020197

**Published:** 2019-02-03

**Authors:** Min Yang, Genli Shen, Mi Liu, Yunfa Chen, Zhen Wang, Qi Wang

**Affiliations:** 1Department of Chemistry, School of Chemistry and Biological Engineering, University of Science and Technology, Beijing 100083, China; yangmin@ustb.edu.cn; 2CAS Key Laboratory of Standardization and Measurement for Nanotechnology, CAS Center for Excellence in Nanaoscience, National Center for Nanoscience and Technology, Beijing 100190, China; shengl@nanoctr.cn (G.S.); liumi@nanoctr.cn (M.L.); 3State Key Laboratory of Multiphase Complex Systems, Institute of Process Engineering, Chinese Academy of Sciences, Beijing 100190, China; yfchen@home.ipe.ac.cn

**Keywords:** CeO_2_-MnO_x_ composite oxides, oxalate method, mesoporous structure, benzene oxidation

## Abstract

The catalytic activities of CeO_2_-MnO_x_ composite oxides synthesized through oxalate method were researched. The results exhibited that the catalytic properties of CeO_2_-MnO_x_ composite oxides were higher than pure CeO_2_ or MnO_x_. When the Ce_at_/Mn_at_ ratio was 3:7, the catalytic activity reached the best. In addition, the activities of CeO_2_-MnO_x_ synthesized through different routes over benzene oxidation were also comparative researched. The result indicated that the catalytic property of sample prepared by oxalate method was better than others, which maybe closely related with their meso-structures. Meanwhile, the effects of synergistic interaction and oxygen species in the samples on the catalytic ability can’t be ignored.

## 1. Introduction

Volatile organic compounds (VOCs) produced by industrial manufacturing are an important class of air pollutants. Among the VOCs, aromatic compounds are one of the major hazardous pollutants, in which benzene is considered to be one of the representing aromatic materials extensively applied in the industry. The complete catalytic oxidation of benzene is often studied as a model reaction, characteristic of the catalytic combustion of VOCs due to its chemical stability [1,2,3,4,5]. The selection of catalysts is important to catalytic degradation of benzene. At present, both classes of catalysts i.e., noble metals and transition metal oxides, have been widely studied for the degradation of VOCs [6,7,8,9]. However, the usage of noble-metal-based catalysts is limited due to high cost, low thermal stability and sensitivity to poisoning. Transition metal oxide-based catalysts are suitable alternative because of higher thermal stability and lower price [10]. In certain cases, transition metal oxides can be actually more active than noble metal catalysts [11].

Ceria (CeO_2_), as a typical rare earth oxide, was investigated in heterogeneous catalysis field due to its high oxygen storage capacity. It can provide active oxygen species to ensure the catalytic reaction. More recently, CeO_2_-based composite oxides were employed for VOCs removal and obtain satisfied results, especially Ce-Mn composites [12,13,14]. CeO_2_-MnO_x_ can be applied as heterogeneous catalysts for the abatement of contaminants in the liquid and gas phases, such as the catalytic reduction of NO and oxidation of acrylic acid and formaldehyde, which exhibit much higher catalytic activity than those of pure MnO_x_ and CeO_2_ [15,16,17,18].

In our previous work [19], Mn element was also doped or mixed with CeO_2_ to obtain Ce-Mn composites through hydrothermal method. Their catalytic behaviours over benzene oxidation were researched, among which, all of Ce-Mn composites exhibited higher activity than MnO_x_ and CeO_2_. However, the best conversion temperature of benzene oxidation over Ce-Mn composite was ca. 375 °C, which could not be applied in the moderate or lower temperature range (100–200 °C) [19], even if supporting noble metal species [20]. Therefore, we need to prepare catalysts with better performance through adjusting microstructure. In this article, we report a series of Ce-Mn composites synthesized through different routes. Their catalytic activities over benzene oxidation are researched comparatively so as to acquire more active catalyst in the temperature range of 100 and 200 °C. Their microstructures are also analyzed in detail so that understanding the elements influencing the activity of samples. 

## 2. Experimental

### 2.1. The Preparation of Ce-Mn Composite Oxides

The chemicals used in this work, including Ce(NO_3_)_3_·6H_2_O (99%), Mn(CH_3_COO)_2_·4H_2_O, Na_2_CO_3_, C_2_H_2_O_4_·2H_2_O, NaOH (98%), and ethanol, were purchased from Beijing Chemicals Company (Beijing, China). A series of Ce-Mn composite oxides were synthesized by carbonate method. Briefly, Ce(NO_3_)_3_·6H_2_O and Mn(CH_3_COO)_2_·4H_2_O in appreciate amounts were dissolved in a 100 mL H_2_O and mixed with a 0.24 M C_2_H_2_O_4_·2H_2_O solution under strong stirring. Then, the mixture was stirred for another 1 h. The precipitates were collected by centrifugation, washed with distilled water and ethanol several times. The obtained materials, labeled as Ce_x_Mn_1−x_ (where x refers to the Ce/(Ce + Mn) atomic ratio) were dried at 80 °C overnight and calcined at 450 ℃ for 2 h with a heating rate of 2 °C·min^−1^. Pure CeO_2_ and MnO_x_ were also prepared using the similar process as reference. In order to compare the catalytic activity, Ce-Mn composite oxides were also synthesized through carbonate method and hydrothermal method [19,21].

### 2.2. Characterization Technique

The crystal phase of the materials was characterized on X-ray diffraction (XRD, Philips, Amsterdam, The Netherlands) equipped with a Cu Kα radiation source (λ = 0.154187 nm) at a scanning rate of 0.03 °/s (2*θ* from 10° to 90°). The assignment of the crystalline phases was based on the ICSD data base (CeO_2_ no. 81-0792; Mn_3_O_4_ no. 80-0382; Mn_2_O_3_ no. 89-4836). The morphology images of catalysts were recorded on a scanning electron microscopy (SEM, JEOL JSM-6700F, Tokyo, Japan) operating at 15 kV and 10 μA. The microstructures of catalysts were examined using transmission electron microscopy (TEM, JEOL JEM-2010F) with an accelerating voltage of 200 kV.

The BET specific surface area (*S*_BET_) was measured by physical adsorption of N_2_ at the liquid nitrogen temperature using an Autosorb-1 analyzer (Quantachrome, Boynton Beach, FL, USA). Before measurement, the samples were degassed at 300 °C for 4 h under vacuum. Surface composition was determined by X-ray photoelectron spectroscopy (XPS, VG Scientific, Waltham, MA, USA) using an ESCALab220i-XL electron spectrometer from VG Scientific with a monochromatic Al Kα radiation. The binding energy (BE) was referenced to the C1s line at 284.8 eV from adventitious carbon.

Hydrogen temperature-programmed reduction (H_2_-TPR) was performed with a U-type quartz reactor equipped with Automated Catalyst Characterization System (Autochem 2920, MICROMERITICS). A 50 mg sample (40–60 mesh) was loaded and pretreated with a 5% O_2_ and 95% He mixture (30 mL/min) at 150 °C for 1 h and cooled to 50 °C under He flow. The samples were then heated to 900 °C at a rate of 10 °C/min under the flow of a 10% H_2_ and 90% Ar mixture (30 mL/min). 

### 2.3. Catalytic Activity Tests

Activity tests for catalytic oxidation of benzene over Ce_x_Mn_1−x_ composite catalysts were performed in a continuous-flow fixed-bed reactor under atmospheric pressure, containing 100 mg of catalyst samples (40–60 mesh). A standard reaction gas containing 1000 ppm benzene and 20% O_2_ in N_2_ was fed with a total flow rate of 100 mL/min. The weight hourly space velocity (WHSV) was typically 60,000 mL·g^−1^·h^−1^. The reactants and the products were analyzed on-line using a GC/MS 6890N gas chromatograph (Hewlett-Packard, Palo Alto, CA, USA) interfaced to a 5973N mass selective detector (Hewlett-Packard, Palo Alto, USA) with a HP-5MS capillary column (30 m × 0.25 mm × 0.25 μm) and another GC (GC112A, Shangfen, Shanghai, China) with a carbon molecule sieve column. The conversion of benzene (*X*_benzene_, %) was calculated as follows:(1)$$X_{\mathrm{benzene}}=\frac{C_{\mathrm{benzene}(\mathrm{in})}-C_{\mathrm{benzene}(\mathrm{out})}}{C_{\mathrm{benzene}(\mathrm{in})}}\times 100\%$$
where, *C*_benzene (in)_ (ppm) and *C*_benzene (out)_ (ppm) are the concentrations of benzene in the inlet and outlet gas, respectively.

## 3. Results and Discussion

### 3.1. Catalytic Oxidation Activity of Ce_x_Mn_1−x_ Composite Oxides for Benzene

The catalytic performance of CeO_2_, MnO_x_ and Ce_x_Mn_1−x_ catalysts was evaluated through the oxidation of benzene. The catalytic conversion of benzene as a function of the temperature, 100–400 °C, is shown in Figure 1a. It can be acquired that MnO_x_ exhibits the least active followed by CeO_2_. With the Mn element adding into CeO_2_, the activity increases monotonically up to a Mn content of 70 at.% and Ce_0.3_Mn_0.7_ is the most active among all catalysts achieving complete benzene conversion at ca. 200 °C. The MnO_x_ and CeO_2_ catalysts synthesized through oxalate route also achieve full conversion at ca. 300 °C. In addition, the activities of Ce_0.3_Mn_0.7_ synthesized through different routes over benzene oxidation are comparative researched in order to identify the advantage of oxalate route (Figure 1b). The result indicates that the sample exhibits higher activity than that of corresponding samples synthesized by hydrothermal or carbonate routes, which is probably related with the microstructure of catalyst. For the purposes of comparison, the reaction temperatures T_10%_, T_50%_, T_90%_ (corresponding to the benzene conversion = 10%, 50%, 90%) used to evaluate the performances of the catalysts are summarized in Table 1.

### 3.2. Characterization of Ce_x_Mn_1−x_ Catalysts

Figure 2a shows the XRD patterns of the samples in the angular range 10–70 2θ. The diffraction peaks at 2θ = 28.5, 33.0, 47.4, 56.4 and 59.2 in the XRD profile of the pure cerium oxide clearly demonstrate the presence of cubic fluorite structure of CeO_2_ (JCPDS 81-0792). For pure MnO_x_, the XRD pattern exhibits complex diffraction peaks which can be considered as a mixture of crystalline Mn_3_O_4_ (JCPDS 80-0382) and Mn_2_O_3_ (JCPDS 89-4836). The data demonstrates that manganese oxide is a multivalent framework manganese (2^+^ and 3^+^), which indicates the existence of various manganese oxides. For Ce_x_Mn_1−x_ catalyst, the XRD patterns exhibit more complex. The XRD patterns of the Ce_x_Mn_1−x_ mixed oxide (x ≥ 0.5) do not show any diffraction of manganese oxides, and only broad reflections attributed to CeO_2_ are observed, which is possible to be related with the formation of solid solution between MnO_x_ and CeO_2_. The diffraction patterns of Ce_x_Mn_1−x_ mixed oxides at x ≤ 0.3 show crystallization of Mn_3_O_4_ and Mn_2_O_3_ except that of CeO_2_. 

The formation of solid solution between MnO_x_ and CeO_2_ can be further evidenced by the fact that the characteristic diffraction peak of CeO_2_ in the composite oxides is slightly shifted to higher values of the Bragg angles, compared with the pure CeO_2_ (Figure 2b). Since the ionic radius of Mn^2+^ (0.091 nm) and Mn^3+^ (0.066 nm) are both smaller than that of the Ce^4+^ (0.1098 nm), the incorporation of Mn^2+^ or Mn^3+^ into the CeO_2_ lattice to form CeO_2_-MnO_x_ solid solution would result in remarkable decrease in the lattice parameter of CeO_2_ in the Ce_x_Mn_1−x_ composite oxide. Meanwhile the O vacancy is also easier to form in order to balance charge due to Ce^4+^ replaced by Mn^2+^ or Mn^3+^ in the Ce_x_Mn_1−x_ catalyst. The oxygen vacancy is beneficial to catalytic activities of Ce_x_Mn_1−x_ catalyst [19]. 

The SEM images of as-prepared Ce_x_Mn_1−x_ oxide catalysts are presented in Figure 3a. For pure oxide CeO_2_, it can be seen that the catalyst is composed of many thin flakes, which are overlapped together to form butterfly-like structure (inset picture). The thickness of every flake is about 200 nm and the length can extend to several micrometers. In the SEM image of MnO_x_, a lot of grains with ellipsoid-like morphology are seen clearly and the size is ca.10 μm. At the surface of every grain, deep ravines are also obviously observed (inset picture). For Ce_x_Mn_1−x_ composite catalysts, the morphology changes gradually with the Mn content increasing. When the Mn theoretical content reaches 50% (Ce_0.5_Mn_0.5_), a few of bulk-like particles with size of several micrometers can be detected except thin flakes. When the theoretical content of Mn ion reaches to 70% (Ce_0.3_Mn_0.7_), a large number of grains possessing layered structure can be observed. In addition, it can be acquired that Ce, Mn and O elements are dispersed together homogeneously through element distribution over Ce_x_Mn_1−x_ composite catalysts. 

Through the discussion over catalytic activities of Ce_x_Mn_1−x_, it has been acquired that Ce_0.3_Mn_0.7_ presents the highest activity and preparation route is also important to the property of catalyst. Therefore, the TEM images of Ce_0.3_Mn_0.7_ prepared through different technology routes are shown so as to analyze their microstructures (Figure 3b). In the TEM image of Ce_0.3_Mn_0.7_ synthesized by oxalate route, the catalyst is composed of thin flakes. At the surface of flake, some mesoporous structures can be observed and the size of mesoporous is ca. 2 nm. As we known, the oxalate chains chelated in the corresponding precursors are easy to be decomposed into CO_x_ and H_2_O with calcination in air, which can leave behind large numbers of voids due to the release of gaseous CO_x_ and H_2_O [21]. Meanwhile, some primary nanoparticles are assembled together thereby forming porous structure, which is beneficial to absorb and desorb the gas due to the formation of massive active sites and decrease of mass transfer effect [22,23]. For Ce_0.3_Mn_0.7_ synthesized by carbonate route, some grains with dumbbell shape can be seen, which are composed of some stacked nanoparticles with the size of 1–2 nm. In the TEM of Ce_0.3_Mn_0.7_ prepared through hydrothermal method, a lot of nanorods with the diameter of ca. 10 nm and the length of 300–400 nm are observed, which is also composed of some assembled nanoparticles. 

In addition, the N_2_ adsorption–desorption isotherms and the pore size distribution of the as-prepared catalysts are displayed in Appendix A. The data show that the isotherms of the as-prepared materials possess type IV characteristics with well-developed H3 type hysteresis loops. The result indicates that the Ce_x_Mn_1−x_ composite catalysts possess porous structure, which is consistent with the result of SEM. The porous structure can facilitate the adsorption and diffusion of reactive molecules, thus greatly reducing limitations of inter-phase mass transfer and enhancing their catalytic activities [21].

The oxidation state of catalyst surface species was examined by XPS analysis. Figure 4 exhibits XPS patterns of Ce 3d, Mn 2p, Mn 3s and O 1s for samples, respectively. In the Ce 3d spectrum of support (Figure 4a), six peaks labeled as V_0_ (882.1 eV), V_1_ (888.7 eV), V_2_ (898.1 eV), V_0_′ (900.7 eV), V_1_′ (907.1 eV) and V_2_′ (916.3 eV) can be identified as characteristic of Ce^4+^ 3d final states [24,25]. The high BE doublet (V_2_/V_2_′) is attributed to the final state of Ce(IV)3d^9^4f^0^O2p^6^, doublet V_1_/V_1_′ is originated from the state of Ce(IV)3d^9^4f^1^O2p^5^, and doublet V_0_/V_0_′ corresponds to the state of Ce(IV)3d^9^4f^2^O2p^4^. The character peaks of Ce^3+^ are also observed at 903.4/884.7 eV and 897.6/879.3 eV labeled as U_1_/U_1_′ and U_0_/U_0_′, respectively [26]. The amount of Ce^3+^ is estimated to be 11.05, 10.89, 10.05 and 5.65% for CeO_2_, Ce_0.7_Mn_0.3_, Ce_0.5_Mn_0.5_ and Ce_0.3_Mn_0.7_, which can be calculated according to the Equation (2). Therefore, Ce species in the Ce_x_Mn_1−x_ composite oxides exist mainly in tetravalent oxidation state.
(2)XCe3+=ACe3+SCe∑A(Ce3++Ce4+)SCe×100%
where $X_{Ce^{3+}}$ is the percentage content of Ce^3+^, *A* is the integrate area of characteristic peak in the XPS pattern, *S* is sensitivity factors (*S* = 7.399).

Figure 4 shows the Mn 2p XPS spectra of Ce_x_Mn_1−x_ composite oxides, in which Mn 2p doublet can be distinguished. The binding energies of the Mn 2p_3/2_ component appear at 641.7 eV and those for Mn 2p_1/2_ appear at 653.2 eV. The BE values of the Mn 2p_3/2_ (641.7 eV) and spin–orbit splitting (11.7 eV) are well matching with the reported values of the trivalent manganese [27]. The shoulder of the Mn 2p_3/2_ component at 640.7 eV is attributed to Mn^2+^ species [28]. The XPS results do not provide any evidence for the presence of Mn^4+^ species (642.2–643 eV) [29,30]. In order to determine the chemical states of Mn further, Mn 3s XPS spectra of Ce_x_Mn_1−x_ are analyzed (Figure 4c). The spin−orbit splitting value (ΔEs) between the two doublets was 5.44 eV for all samples, closing to the value of 5.1 for the standard sample of α-Mn_2_O_3_. The ΔEs of MnO is about 6.3 eV, indicating that the oxidation status of Mn is predominantly tervalent [31,32]. The average oxidation state of Mn was calculated using formula A_OS_ = 8.95 − 1.13 × ΔEs [33]. The data was calculated to be 2.80 that fall in between the average oxidation state of Mn (+2.67) in Mn_3_O_4_ and the state of Mn (+3) in Mn_2_O_3_. Therefore, the element Mn in the Ce_x_Mn_1−x_ catalysts is existed in the form of Mn_3_O_4_ and Mn_2_O_3_, which is consistent with the data of XRD. In addition, the peak of Mn 2p3/2 (641.5 eV) for Ce_x_Mn_1−x_ composites is shifted to lower binding energy comparing with pure MnO_x_, which may be a consequence of interaction between CeO_2_ and MnO_x_ [19]. It is worthy to note that the pattern and the intensity of Ce_0.7_Mn_0.3_ is similar and less weaker compared with other Ce_x_Mn_1−x_ composites. The corresponding XPS spectrum can be seen in the Appendix A.

The XPS O1s spectra (Figure 4d) show a main peak at a binding energy of 529.1–529.9 eV, corresponding to lattice oxygen of CeO_2_ and MnO_x_ phases(O^2−^; denoted as O_α_) [27,30]. A broad shoulder at the higher binding energy region (531.3–531.8 eV) is ascribed to defective oxides or oxygen species of the surface carbonates and hydroxide (denoted as O_β_) [18,34]. It is worthy to note that the peak corresponding to lattice oxygen in Ce_x_Mn_1−x_ composite catalyst with higher Mn content tends to shift toward higher BE value than that of pure CeO_2_ (from 529.1 eV to 529.6 eV), which suggests that the environments of oxygen change with increasing Mn content. This appearance is also attributed to the interaction between CeO_2_ and MnO_x_. In addition, the content of O_α_ is calculated according to the Equation (3) and listed in Table 2. The data shows that the Ce_0.3_Mn_0.7_ sample possesses more lattice oxygen species, confirming that the mobility and availability of lattice oxygen species are enhanced due to the synergistic effects of CeO_2_ and MnO_x_ in Ce_0.3_Mn_0.7_ [35].
(3)XOα=AOαSO∑A(Oα+Oβ)SO×100%
where $X_{O_{\alpha }}$ is the percentage content of O_α_, *A* is the integrate area of characteristic peak in the XPS pattern, *S* is sensitivity factors (*S* = 0.711).

In order to check the redox properties of the new series of Ce_x_Mn_1−x_ systems, TPR of all the catalysts were carried out. Figure 5a shows the H_2_-TPR profiles of CeO_2_, MnO_x_ and Ce_x_Mn_1−x_ composite oxides. Similar to previous findings [36,37,38], pure CeO_2_ exhibits two reduction peaks at around 405 °C and about 719 °C. The former low-temperature reduction is due to the removal of surface oxygen and the later high-temperature reduction is related to the oxygen species in bulk CeO_2_. The H_2_-TPR profile of pure MnO_x_ shows two strong reduction peaks at 167 °C and 330 °C, respectively, with an area ratio of the lower to the higher temperature hydrogen consumption of about 1:2.42. The actual hydrogen consumption of two reduction peaks is 0.92343 and 1.88118 mmol/g and the corresponding ratio is 1:2.04. As we known, Mn_2_O_3_ proceed two-step reduction, the low-temperature reduction peak represented the reduction of Mn_2_O_3_ to Mn_3_O_4_ and the high-temperature reduction peak referred to the further reduction of Mn_3_O_4_ to MnO [39]. The theoretical hydrogen consumption ratio is 1:2 as seen in Formula (4) and (5), which is less than the fitted value (2.42) and actual data (2.04). It indicates that the hydrogen gas is more consumed at higher temperature, which is attributed to the extra existence of Mn_3_O_4_ phase. This is in agreement with the XRD data, which show that the crystalline phase of pure MnO_x_ corresponds to Mn_2_O_3_ and Mn_3_O_4_.
(4)$$3Mn_{2}O_{3}+H_{2}\to 2Mn_{3}O_{4}+H_{2}O$$
(5)$$Mn_{3}O_{4}+H_{2}\to 3MnO+H_{2}O$$

In contrast to pure CeO_2_ and MnO_x_, the reduction profiles of Ce_x_Mn_1−x_ catalysts are more complicated. For Ce_x_Mn_1−x_, the TPR profiles consist of three overlapping peaks at lower temperature and one peak at higher temperature. According to the reduction characteristics of pure MnO_x_ and CeO_2_, it can be deduced that the lower temperature peaks (182/351 °C, 189/340 °C, 169/343 °C) are assigned to the two-step reduction of Mn_2_O_3_. The peaks at 693 °C or 714 °C are attributed to the oxygen species in bulk CeO_2_. It is worthy to note that another obvious peak at lower temperature (295 °C, 283 °C or 232 °C) as shown in Figure 5b is possible to be caused by the synergistic effect between Mn^2+^/Mn^3+^ and Ce^4+^, which is related with CeO_2_-MnO_x_ solid solution. It can facilitate the mobility of the oxygen species in the Ce_x_Mn_1−x_ composite oxide. Therefore, their catalytic activities over benzene are higher than pure CeO_2_ and MnO_x_. Additionally, the reduction temperature is lower and the peak corresponding to the oxygen species in bulk CeO_2_ disappear in the TPR pattern of Ce_0.3_Mn_0.7_ compared with those of Ce_0.7_Mn_0.3_/Ce_0.5_Mn_0.5_. This indicates that the reduction of the manganese oxide and the cerium oxide in Ce_0.3_Mn_0.7_ is promoted more obvious, which results in the highest activity over benzene in the Ce_x_Mn_1−x_ composite oxides. In the view of hydrogen consumption, the same conclusion can be also obtained. Ce_0.3_Mn_0.7_ consumed the most hydrogen gas (4.88447 mmol/g) among Ce_x_Mn_1−x_ composite oxides indicating that Ce_0.3_Mn_0.7_ possesses more oxygen species, which are beneficial to benzene oxidation reaction. Therefore, Ce_0.3_Mn_0.7_ presents higher activity compared with other Ce_x_Mn_1−x_.

### 3.3. Factors Influencing the Catalytic Activity

Through the analysis, it has been acquired that Ce_0.3_Mn_0.7_ performs excellently in catalytic oxidation of benzene among the Ce_x_Mn_1−x_ catalysts. Many factors are considered to determine catalytic performances. Firstly, the existence of Ce-Mn solid solution results in the formation of oxygen vacancies due to incorporate Mn into CeO_2_ crystal lattice, which can enhance their activities. Secondly, the large numbers of active sites will be introduced after removal of oxalate chains, because of the formation of porous structure which can also facilitate the adsorption and diffusion of organic molecules, thus reducing limitations of interphase mass transfer and promoting their catalytic activities. Thirdly, the better reducibility at low temperature also plays a great role in the catalytic activity. Through the analysis of TPR, it has been acquired that the reductions of Ce_x_Mn_1−x_ catalysts start at the lower temperature compared with CeO_2_, which indicate that the catalysts possess more highly reducible surface species such as absorbed oxygen. Additionally, the existence of special peak caused by the strong interaction between Ce and Mn compared with MnO_x_ also result in the enhancement of reducibility. Finally, the oxidation of organic molecules over transition metal oxide or mixed metal oxide catalysts involves two identical mechanisms: a Langmuir–Hinshelwood mechanism at lower temperature and a Mars–van Krevelen mechanism with increasing reaction temperature [28,40]. At lower temperature, the adsorbed oxygen species with higher activity can enhance the adsorption and oxidation of VOCs. With the temperature rising, the adsorbed organic molecules are oxidized by the oxygen of metal oxides, which can be replenished by gas phase oxygen. Therefore, the adsorbed oxygen species will have an important role to play in determining its catalytic activity. As displayed in Table 2, the Ce_0.3_Mn_0.7_ exhibits a higher content of Ce^4+^ and Mn^3+^. The high-valence of cerium and manganese ions are preferred to adsorb more active oxygen species to attend in the reaction, thereby Ce_0.3_Mn_0.7_ possessed higher activity. 

## 4. Conclusions

A series of Ce_x_Mn_1−x_ composite oxides, CeO_2_ and MnO_x_ were synthesized through oxalate method and the complete catalytic oxidation of benzene were examined. The results indicated that Ce_x_Mn_1−x_ catalysts exhibited better activities comparing with pure CeO_2_ or MnO_x_, among which the catalytic activity reached the best when the Ce_at_/Mn_at_ optimum ratio was 3:7. In order to identify the advantage of oxalate route, Ce-Mn composite oxides were also synthesized through carbonate method and hydrothermal method. The results indicated that the samples prepared by oxalate route exhibited higher activities, which were probably related with the microstructure of catalyst. Additionally, the influence of oxygen vacancy and synergistic effect in the benzene catalytic oxidation can’t be also ignored.

## Figures and Tables

**Figure 1 nanomaterials-09-00197-f001:**
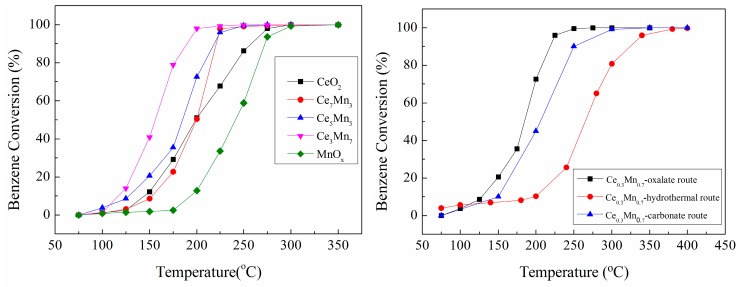
C_6_H_6_ conversion (%, (**a**)) over CeO_2_, MnO_x_ and Ce_x_Mn_1−x_ catalysts synthesized by oxalate method; C_6_H_6_ conversion comparison as a function of reaction temperature over Ce_3_Mn_7_ through different reaction route (**b**).

**Figure 2 nanomaterials-09-00197-f002:**
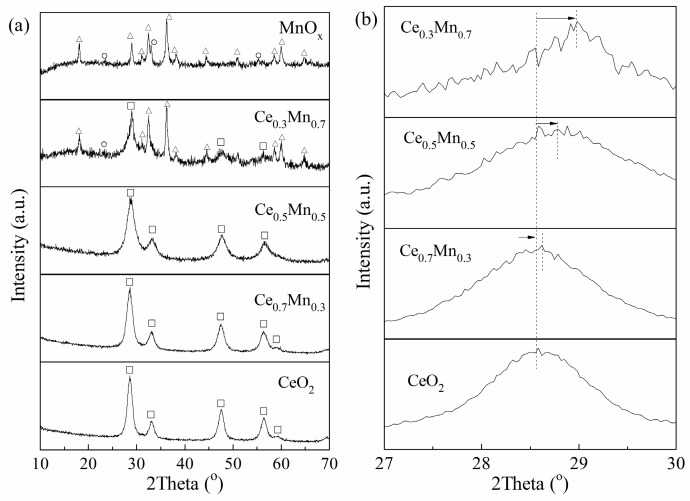
XRD patterns of all the samples: (**a**) wild angle patterns, and (**b**) Enlarged-zone patterns. Crystalline phases detected: (o) CeO_2_, (□) Mn_3_O_4_, (∆) Mn_2_O_3_.

**Figure 3 nanomaterials-09-00197-f003:**
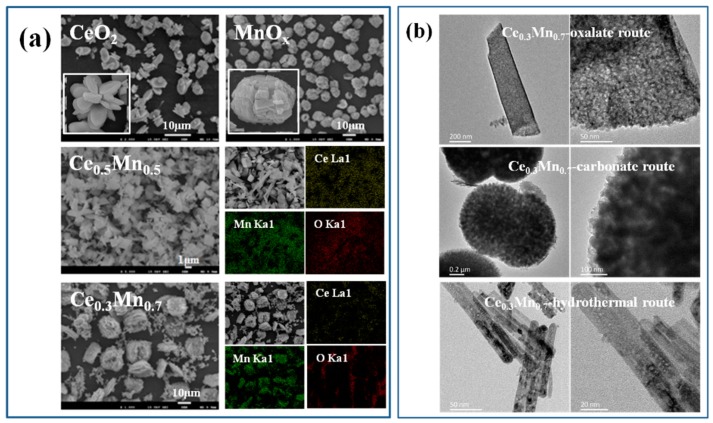
SEM images and EDX mapping of CeO_2_, MnO_x_, Ce_0.5_Mn_0.5_ and Ce_0.3_Mn_0.7_ (**a**); TEM images of Ce_0.3_Mn_0.7_ synthesized through different routes (**b**).

**Figure 4 nanomaterials-09-00197-f004:**
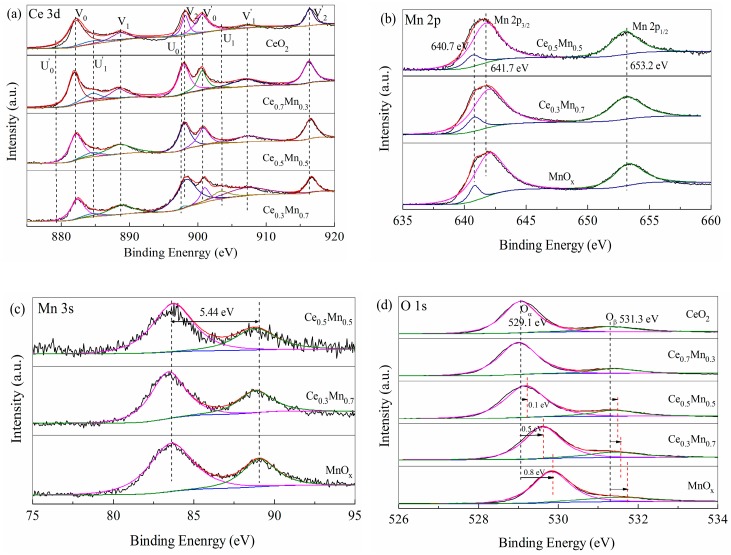
X-ray photoelectron spectra in the Ce 3d (**a**), Mn 2p (**b**), Mn 3s (**c**), O 1s (**d**) regions for the Ce_x_Mn_1−x_ catalysts.

**Figure 5 nanomaterials-09-00197-f005:**
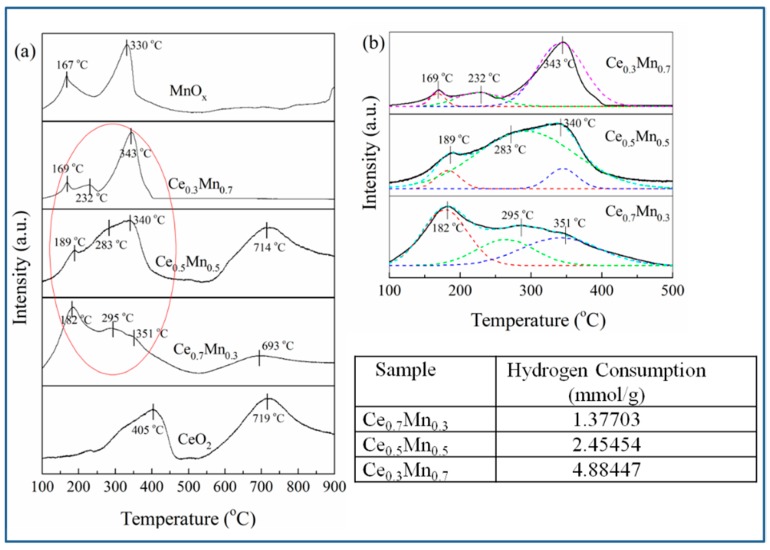
H_2_-TPR of CeO_2_, MnO_x_, Ce_x_Mn_1−x_ (**a**) and Enlarged picture of the red circle area (**b**).

**Table 1 nanomaterials-09-00197-t001:** Catalytic activities of CeO_2_, MnO_x_ and Ce_x_Mn_1−x_ catalysts.

Catalyst	Benzene Oxidation Activity		
	T_10%_ (°C)	T_50%_ (°C)	T_90%_ (°C)
CeO_2_	145	200	257
Ce_0.7_Mn_0.3_	153	200	220
Ce_0.5_Mn_0.5_	128	185	218
Ce_0.3_Mn_0.7_	117	156	190
MnO_x_	194	241	273
Ce_0.3_Mn_0.7_ (Hydrothermal)	200	265	350
Ce_0.3_Mn_0.7_ (Carbonate)	150	220	250

**Table 2 nanomaterials-09-00197-t002:** XPS results of Ce_x_Mn_1−x_ samples.

Sample	O		O_α_/(O_α_ + O_β_) (%)	Ce (%)	Mn (%)
	O_α_	O_β_		Ce^3+^/(Ce^3+^ + Ce^4+^)	Mn^2+^/(Mn^2+^ + Mn^3+^)
CeO_2_	529.0	531.3	77.16	11.05	-
Ce_0.7_Mn_0.3_	529.1	531.3	81.01	10.89	16.23
Ce_0.5_Mn_0.5_	529.1	531.3	82.04	10.05	15.12
Ce_0.3_Mn_0.7_	529.2	531.5	83.18	5.65	13.42
MnO_x_	529.6	531.6	78.47	-	10.69

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
