# Peer review of "Preparation of Ce–Mn Composite Oxides with Enhanced Catalytic Activity for Removal of Benzene through Oxalate Method"

_nanomaterials, 2019, doi:10.3390/nano9020197_

Reviewer 1 Report

The paper presents results for the oxidation of benzene over CeO2-MnOx catalysts.  The catalysts were characterized by SEM, XRD, BET, TPR, XPS, and TEM.  The authors found that a 3:7 atomic ratio of Ce:Mn was optimal.

The English needs to be improved.

The major criticism has to do with the analysis of the XPS results.  The paper indicates that the percentage of alpha O is important for activity and that higher is better.  The data in this paper supports that.  However, one reads reference 19 where they did a similar study, the catalysts in 19 are not as good as those in this paper yet they have higher percentages of the alpha O.  This should be discussed in the paper if there is a reasonable explanation.

The description of the MnOx TPR results is confusing.  One expects a certain ratio of H2 consumed for Mn2O3 to Mn3O4 compared to Mn3O4 to MnO.  However these results suggest that there is extra Mn3O4 present.  Could it simply be that the synthesized catalyst does not start as pure Mn2O3 rather it is a mixture of Mn2O3 and Mn3O4?  If this is what the authors are saying, they should clear up line 244-245.

Overall, the paper presents interesting results on a system that is only slightly different than that previously published.  The results are similar with respect to the Ce:Mn ratio.  However, there seems to be some inconsistency between the current paper and ref 19, which puts the results in question.

Author Response

Reviewer #1:

1. The major criticism has to do with the analysis of the XPS results. The paper indicates that the percentage of alpha O is important for activity and that higher is better. The data in this paper supports that.  However, one reads reference 19 where they did a similar study, the catalysts in 19 is not as good as those in this paper yet they have higher percentages of the alpha O. This should be discussed in the paper if there is a reasonable explanation.

Answer: Thank the reviewer for the comments. The suggestion is important to us. The content of Oα is important to the catalytic activity, however it is not only element in determining the catalytic property. In this article, the synthesized catalysts possess meso-structure and higher surface area, which can facilitate the adsorption and diffusion of reactive molecules, meanwhile the limitations of inter-phase mass transfer can be reduced. Therefore, their catalytic activities are enhanced.

 2. The description of the MnOx TPR results is confusing. One expects a certain ratio of H2 consumed for Mn2O3 to Mn3O4 compared to Mn3O4 to MnO. However these results suggest that there is extra Mn3O4 present. Could it simply be that the synthesized catalyst does not start as pure Mn2O3 rather it is a mixture of Mn2O3 and Mn3O4? If this is what the authors are saying, they should clear up line 244-245.

Answer: Thank the reviewer for the comments. The description of the MnOx TPR results have be revised according to the reviewer’s suggestion

Reviewer 2 Report

The manuscript titled “Preparation of Ce–Mn composite oxides with enhanced catalytic activity for removal of benzene through oxalate method” evaluates the removal of benzene by oxidation. In my opinion, the manuscript is interesting although the authors should highlight the novelty of the results.

With regard to the catalytic data, the authors should clarify what are the benefits of this synthetic process. In my opinion, the authors should synthesize another Ce-Mn catalyst or compare the obtained data with those reported in the literature.

Maybe, it could be interesting recover the catalyst after the reaction and study the evolution of the active phase.

With regard to the XRD data, I think that it is interesting to evaluate the crystal size of the catalysts.

In the N2 adsorption-desorption, I don’t know if the adsorption isotherm are IV or II. The presence of interparticle voids favors the N2-filling at higher relative pressure.

In the XPS, the authors could compare the theoretical Mn/Ce molar ratio with those obtained on its surface.

On the other hand, the authors should consider that Ce is photo-reducible easily so it is necessary short irradiation times. The presence of Ce3+ can be increase by this fact.

Author Response

Reviewer #2:

1. With regard to the catalytic data, the authors should clarify what are the benefits of this synthetic process. In my opinion, the authors should synthesize another Ce-Mn catalyst or compare the obtained data with those reported in the literature.

Answer: Thank the reviewer for the comments. In this article, the Ce-Mn catalysts with the same Ce/Mn ratio synthesized through other routes have been reported. The catalytic activities have been also comparatively researched. The results exhibited that Ce-Mn catalysts prepared through oxalate method possessed the best activity. The oxalate method can be operated at room temperature compared with hydrothermal route, and the catalyst prepared possesses meso-porous structure, which is beneficial to the catalytic reaction.

 2. Maybe, it could be interesting recover the catalyst after the reaction and study the evolution of the active phase.

Answer: Thank the reviewer for the comments. In the following work, the evolution of the active phase after the reaction will be discussed in detail.

 3. With regard to the XRD data, I think that it is interesting to evaluate the crystal size of the catalysts.

Answer: Thank the reviewer for the comments. The suggestion is important to us. The crystal sizes of the catalysts have been calculated according to Scherrer equation. The sequence of crystal size is followed by CeO2(10.3nm)0.7Mn0.3(13.4nm)0.5Mn0.5 (24.2nm)Ce0.3Mn0.7(25.3nm)x(40.3nm).

 4. In the N2 adsorption-desorption, I don’t know if the adsorption isotherm are IV or II. The presence of inter-particle voids favors the N2-filling at higher relative pressure.

Answer: In the manuscript, the N2 adsorption–desorption isotherms of the as-prepared catalysts have been displayed in Supporting Information. The adsorption isotherm is attributed to IV.

 5. In the XPS, the authors could compare the theoretical Mn/Ce molar ratio with those obtained on its surface.

Answer: Thank the reviewer for the comments. The reviewer’s suggestion is useful. At present, the ICP results of Ce-Mn composites have not been obtained. In the following work, we will supplement the comparative result of the theoretical Mn/Ce molar ratio and those obtained on its surface.

Reviewer 3 Report

I find the paper very nice, with interesting results and with extensive characterisation done. However, the English, although understandable, needs a moderate revision, preferentially by a native english speaker. Authors should better explain, in the text, abstract and conclusions the possible reaons why the Ce: Mn ration of 3:7 gave the bext results. 

Also the paper should be re-estructured. It is easier if characterisation results are presented first and only then the catalytic results, instead of the opposite.

Author Response

Reviewer #3:

1. However, the English, although understandable, needs a moderate revision, preferentially by a native English speaker. Authors should better explain, in the text, abstract and conclusions the possible reasons why the Ce: Mn ration of 3:7 gave the best results.

Answer: We have invited some experienced colleagues to help with the revision of English throughout the manuscript to make that meets the journal’s desired standard. The XPS analysis of Ce0.3Mn0.7 exhibited that the catalyst possessed more lattice oxygen species and higher content of Ce4+ and Mn3+, which are preferred to adsorb more active oxygen species to attend in the reaction. The TPR description of Ce0.3Mn0.7 showed that the typical reduction temperature was lower compared with those of Ce-Mn catalysts. In the view of hydrogen consumption, Ce0.3Mn0.7 consumed the most hydrogen gas indicating that Ce0.3Mn0.7 possesses more oxygen species, which are beneficial to benzene oxidation reaction. Additionally, the peak caused by the synergistic effect between Mn2+/Mn3+ and Ce4+ exhibited at lower temperature in the TPR pattern of Ce0.3Mn0.7 compared with those of other Ce-Mn catalysts, which also identified the enhancement of redox property. The reviewer’s suggestion is very important to us. In the following work, more characterization results are adopted to explain the possible reasons in detail.

 2. The paper should be restructured. It is easier if characterization results are presented first and only then the catalytic results, instead of the opposite.

Answer: Thank the reviewer for the comments. In my opinion, it is better to describe the catalytic results firstly. It can be obtained that which catalyst exhibits the best activity through catalytic examination. In the following text, all characterization results are used to explain the differences in activity. Therefore, I think that it's better not to change the structure of the article.

Round  2

Reviewer 1 Report

The XPS results are still not consistent with the other paper.  If the authors were to combine the catalysts in the two papers and write a paper about it, the conclusions about the XPS results would be different.  Granted there are other things to consider, one cannot use that to ignore previous data.  Are there differences in how the XPS data was obtained in the other paper??

Author Response

Reviewer #1:

The XPS results are still not consistent with the other paper. If the authors were to combine the catalysts in the two papers and write a paper about it, the conclusions about the XPS results would be different. Granted there are other things to consider, one cannot use that to ignore previous data. Are there differences in how the XPS data was obtained in the other paper?

Answer: Thank the reviewer for the comments. In this article, the Ce-Mn composite catalysts were synthesized through oxalate method at normal atmospheric temperature. In other paper (Ref. 19), the Ce-Mn composite catalysts were synthesized through hydrothermal route. Due to the particularity of hydrothermal method, the crystal structure of catalyst is more complete and the oxidation state of element shows high valence state. Therefore, the oxidation state of Mn element for two types of Ce-Mn composite catalysts exist differences although the patterns of Ce3d and O1s are similar. The Mn element in Ce-Mn composite prepared by hydrothermal method exhibits trivalent, which is consistent with XRD result. The Mn element in Ce-Mn composite synthesized through oxalate method possesses divalent and trivalent, which is also identified by the XRD data.